# Best-BRA (Is subpectoral or prepectoral implant placement best in immediate breast reconstruction?): a protocol for a pilot randomised controlled trial of subpectoral versus prepectoral immediate implant-based breast reconstruction in women following mastectomy

Kirsty Roberts [1], Nicola Mills,[1] Chris Metcalfe [1], Athene Lane,[1] Clare Clement [1], William Hollingworth,[1] Jodi Taylor,[1] Chris Holcombe,[2] Joanna Skillman,[3] Katherine Fairhurst,[1] Lisa Whisker,[4] Ramsey Cutress,[5] Steven Thrush,[6] Patricia Fairbrother,[7] Shelley Potter [1,8]

**Correspondence to**
Shelley Potter;
shelley.potter@bristol.ac.uk

## ABSTRACT

**Background** Implant-based breast reconstruction (IBBR) is the most commonly performed reconstructive procedure following mastectomy. IBBR techniques are evolving rapidly, with mesh-assisted subpectoral reconstruction becoming the standard of care and more recently, prepectoral techniques being introduced. These muscle-sparing techniques may reduce postoperative pain, avoid implant animation and improve cosmetic outcomes and have been widely adopted into practice. Although small observational studies have failed to demonstrate any differences in the clinical or patient-reported outcomes of prepectoral or subpectoral reconstruction, high-quality comparative evidence of clinical or cost-effectiveness is lacking. A well-designed, adequately powered randomised controlled trial (RCT) is needed to compare the techniques, but breast reconstruction RCTs are challenging. We, therefore, aim to undertake an external pilot RCT (Best-BRA) with an embedded QuinteT Recruitment Intervention (QRI) to determine the feasibility of undertaking a trial comparing prepectoral and subpectoral techniques.

**Methods and analysis** Best-BRA is a pragmatic, two-arm, external pilot RCT with an embedded QRI and economic scoping for resource use. Women who require a mastectomy for either breast cancer or risk reduction, elect to have an IBBR and are considered suitable for both prepectoral and subpectoral reconstruction will be recruited and randomised 1:1 between the techniques. The QRI will be implemented in two phases: phase 1, in which sources of recruitment difficulties are rapidly investigated to inform the delivery in phase 2 of tailored interventions to optimise recruitment of patients. Primary outcomes will be (1) recruitment of patients, (2) adherence to trial allocation and (3) outcome completion rates. Outcomes will be reviewed at 12 months to determine the feasibility of a definitive trial.

### Strengths and limitations of this study

► This external pilot randomised controlled trial (RCT) with an embedded QuinteT Recruitment Intervention will determine whether it is possible to recruit and randomise patients to a pragmatic trial comparing prepectoral and subpectoral approaches for implant-based breast reconstruction.

► The QuinteT Recruitment Intervention will allow recruitment challenges to be identified, understood and addressed in real time, allowing a rapid decision about the feasibility of a definitive trial.

► The external pilot RCT will only address the feasibility of recruitment within an implant reconstruction study; a full-scale main study will still be necessary to compare the clinical and cost-effectiveness of prepectoral and subpectoral implant-based breast reconstruction.

**Ethics and dissemination** The study has been approved by the National Health Service (NHS) Wales REC 6 (20/WA/0338). Findings will be presented at conferences and in peer-reviewed journals.

**Trial registration number** ISRCTN10081873.

## INTRODUCTION

Fifty-five thousand women per year in the UK are diagnosed with breast cancer[1] of whom 40% will require a mastectomy.[2] The loss of a breast may dramatically impact women's quality of life[3] and in the UK, the National Institute for Health and Care Excellence recommends offering breast reconstruction to improve outcomes.[4]

Implant-based breast reconstruction (IBBR) is the most commonly performed breast reconstruction procedure in the UK accounting for almost 70% of all immediate reconstructions performed following mastectomy.[5] The introduction of biological or synthetic mesh over the last 10 years has had a major impact on the practice of IBBR. Initially, mesh was sutured between the lower border of the pectoralis muscle and the chest wall extending the subpectoral pocket.[6] This allowed a definitive fixed-volume implant to be placed under the muscle at the time of surgery, avoiding the need for tissue expansion and a second procedure and resulting in better cosmetic outcomes through improved lower pole projection.[7–14] There are, however, limited data to support the safety or effectiveness of mesh-assisted subpectoral techniques.[15–19] The lack of high-quality evidence to support practice is highlighted in the recently updated UK mesh-assisted breast reconstruction guidelines[20] and in March 2021, the United States Food and Drug Administration issued a Safety Communication stating that acellular dermal matrices, the most commonly used form of biological mesh, are not licenced for use in breast reconstruction and recommended careful discussion of the risks and benefits of mesh with women considering surgery.[21] Despite this mesh-assisted procedures have become established as the standard of care in the UK, with two-thirds of the 2108 patients undergoing IBBR between 2014 and 2016 in the UK iBRA multicentre prospective cohort study having mesh.[18] More recently, prepectoral techniques have been introduced in which the implant, fully or partially wrapped in mesh, is placed on top of rather than under the pectoralis major muscle.[22] These 'muscle-sparing' techniques may be less painful and avoid the potentially distressing implant 'animation' seen when the pectoralis muscle contracts.[23]

Due to the perceived advantages of the procedure, prepectoral reconstruction is being widely adopted into practice without high-quality evidence to support its use. Early results are promising,[24–31] but caution is required as subcutaneous implant reconstruction without mesh was previously abandoned by the reconstructive community due to high complication rates.[32–35] Few studies have directly compared the outcomes of prepectoral and subpectoral mesh-assisted techniques.[18 36–40] These are small, often single centre studies with limited follow-up, but have demonstrated no differences in short-term clinical[36–39] or patient-reported[37] outcomes between prepectoral and subpectoral implant placement. The iBRA prospective multicentre cohort study demonstrated that the short-term complications of prepectoral and subpectoral reconstruction were equivalent[18] but suggested that at 18 months, satisfaction with outcome, assessed using the validated BREAST-Q, may be greater in patients receiving prepectoral compared with subpectoral techniques.[19] Caution is required, however as the numbers of patients undergoing prepectoral reconstruction in the iBRA study were small.

High-quality evidence is lacking and there is therefore a need for a well-designed, adequately powered randomised controlled trial (RCT) to evaluate the clinical and cost-effectiveness of IBBR techniques. RCTs in breast reconstruction however are challenging due to patient[41] and surgeon preference[42] and expert opinion suggesting that RCTs in breast reconstruction would be 'unethical', 'impractical' and/or 'inappropriate'.[3 43–46] Previous trials in breast reconstruction have been unsuccessful due to failure to recruit.[41 47] Careful feasibility work to inform the design and conduct of a future study is therefore required before a definitive trial can be planned.

The UK iBRA study[48] used mixed methods including a national practice questionnaire,[49 50] a randomisation acceptability survey[51] and semistructured interviews[52] with key stakeholders to explore the most appropriate design for a future RCT. This work suggested that a well-designed trial would be feasible and that key areas of uncertainty included the use of biological versus synthetic mesh and prepectoral versus subpectoral implant positioning.[51] Prepectoral techniques, however, only emerged during the iBRA study and have gained popularity since with two-thirds of survey respondents now offering the technique. Ongoing work in the Pre-BRA study, an IDEAL 2a/2b study evaluating the safety and stability of prepectoral reconstruction prior to definitive evaluation suggests that the technique is safe and sufficiently stable for evaluation in a clinical trial.[53]

While the majority of respondents felt that an appropriately designed RCT in IBBR may be possible,[51] qualitative interviews with key stakeholders identified a number of barriers to the successful conduct of a future trial.[52] These included limited understanding of a pragmatic study design and the role of randomisation in minimising bias, issues around clinician equipoise and aspects of surgical culture not being supportive of RCTs.

Although the iBRA study addressed many of the feasibility issues relating to a future trial including the most appropriate comparators, it did not include a randomised study. It is therefore not clear whether surgeons will recruit patients to an RCT comparing two approaches to IBBR or whether patients will consent to be randomised and agree to receive their allocated treatment. Further feasibility work is therefore needed.

We therefore aimed to undertake an external pilot RCT (Best-BRA study- is subpectoral or prepectoral implant placement Best in immediate BReAst reconstruction?). In anticipation of recruitment challenges a QuinteT Recruitment Intervention (QRI)—a complex intervention that aims to rapidly identify, understand and address sources of recruitment difficulties[54]—has been embedded in the study protocol. Having previously been applied to over 60 RCTs to date, the QRI has led to insights about recruitment issues and the development of targeted strategies that have facilitated successful completion of surgical trials that had previously been considered impossible.[54 55]

The Best-BRA study will determine whether it is possible to recruit and randomise women to a study comparing prepectoral and subpectoral IBBR and determine the feasibility of progression to a definitive large-scale pragmatic trial.

## METHODS

This protocol adheres to the Standard Protocol Items for Randomised Trials[56] guidelines.

### Study design

The study will consist of a pragmatic multicentre external pilot RCT with an embedded QRI to determine the feasibility of recruitment and randomisation to either a subpectoral or prepectoral implant reconstruction following mastectomy.

### Study setting

NHS breast and plastic surgery units in the UK offering both subpectoral and prepectoral immediate IBBR to women following mastectomy.

### Recruitment

Participants will be recruited and randomised over a 12–18 month period by surgeons and research teams working together at participating centres. All female patients undergoing mastectomy who elect to have an IBBR will form the target population. Research teams at each site will maintain a trial screening log following the Screened, Eligible, Approached, Randomised (SEAR) framework[57] to determine the proportion of:

1. Patients considered eligible for the study (with reasons for non-eligibility).
2. Eligible patients approached about the study (with reasons for not being approached).
3. Eligible patients randomised (with reasons for not being randomised).
4. Randomised patients who receive their allocated treatment (with reasons for not receiving allocated treatments).

The screening log will be reviewed monthly to provide feedback to recruiters and aid understanding of surgeons' and patients' preferences for types of surgery as part of the QRI.

Potential participants who will have a mastectomy for breast cancer or risk reduction will be identified from breast cancer and oncoplastic multidisciplinary team meetings (figure 1). The study will be introduced by the treating surgeon when reconstructive options are discussed, and patients will be given written information outlining the study (online supplemental appendix 1). As a key objective of the study is to understand recruitment challenges, these initial consultations will be recorded with patient consent.

Women will provide written consent prior to study participation (online supplemental appendix 2). This will include optional consent to be contacted for future studies, and consent to assess long-term patient reported outcomes through linkage to routine data sources.

### Eligibility and allocation

#### Inclusion criteria

Patients will be included for the trial if they meet all of the following criteria:

1. They are aged 18 years or above.
2. They require a mastectomy for breast cancer or risk reduction.

3. They elect to undergo immediate IBBR.
4. They are considered eligible for both prepectoral and subpectoral reconstruction by the surgical team.

#### Exclusion criteria

Patients will be excluded from the trial if they meet any of the following criteria:

1. They have delayed reconstruction.
2. They are having revision breast reconstruction surgery.

Patients who smoke, have high body mass index or have had previous radiotherapy to the breast will not be excluded for participation in the trial but surgeons will be encouraged to offer mesh-assisted IBBR in line with updated joint Association of Breast Surgery and British Association of Plastic, Reconstructive and Aesthetic Surgeons' guidelines for the use of mesh in reconstructive procedures.

Participants will be randomised by the local research team after eligibility and consent have been confirmed at the final clinic visit by the treating surgeon prior to admission for surgery. The randomisation sequence will be generated by a statistician independent of participant recruitment using the random number generator in Stata statistical software (V.16, StataCorp, 2019). Patients will be randomly allocated to the techniques in a 1:1 ratio to either subpectoral or prepectoral immediate IBBR stratified by hospital. Women undergoing bilateral surgery will receive the same procedure on both sides. Allocation will be concealed until the patient has been logged into the system and a study identification (ID) number generated so ensuring that judgements about eligibility are made without knowledge of the next allocation.

### Intervention

All patients will undergo a skin or nipple-preserving, or skin-reducing mastectomy followed by an IBBR. Participating surgeons will undertake the procedure as per their standard practice. Mesh choice (biological or synthetic and the product used) and implant selection (fixed volume; adjustable implants or tissue expanders) will be as per surgeon preference. Two surgeon/two team operating (both procedures performed simultaneously) will be encouraged in bilateral cases to minimise operative time. The following steps of the IBBR procedure will be considered mandatory, prohibited and flexible/discretionary according to the typology described by Blencowe et al[58]:

► Mandatory: insertion of a tissue expander/adjustable implant or fixed volume implant.
► Prohibited: raising the pectoralis muscle if allocated to prepectoral trial arm.
► Flexible/discretionary: all other steps of the procedure.

### Standard care during and post procedure

Strategies to minimise infection (eg, use of laminar flow, cavity irrigation, glove change) will be as per local practice but participating centres will be encouraged to adhere to best practice[59] and use the evidence-based Manchester Theatre Implant Checklist (TIC) wherever possible.[60] Use of drains and other concomitant interventions will be permitted as per

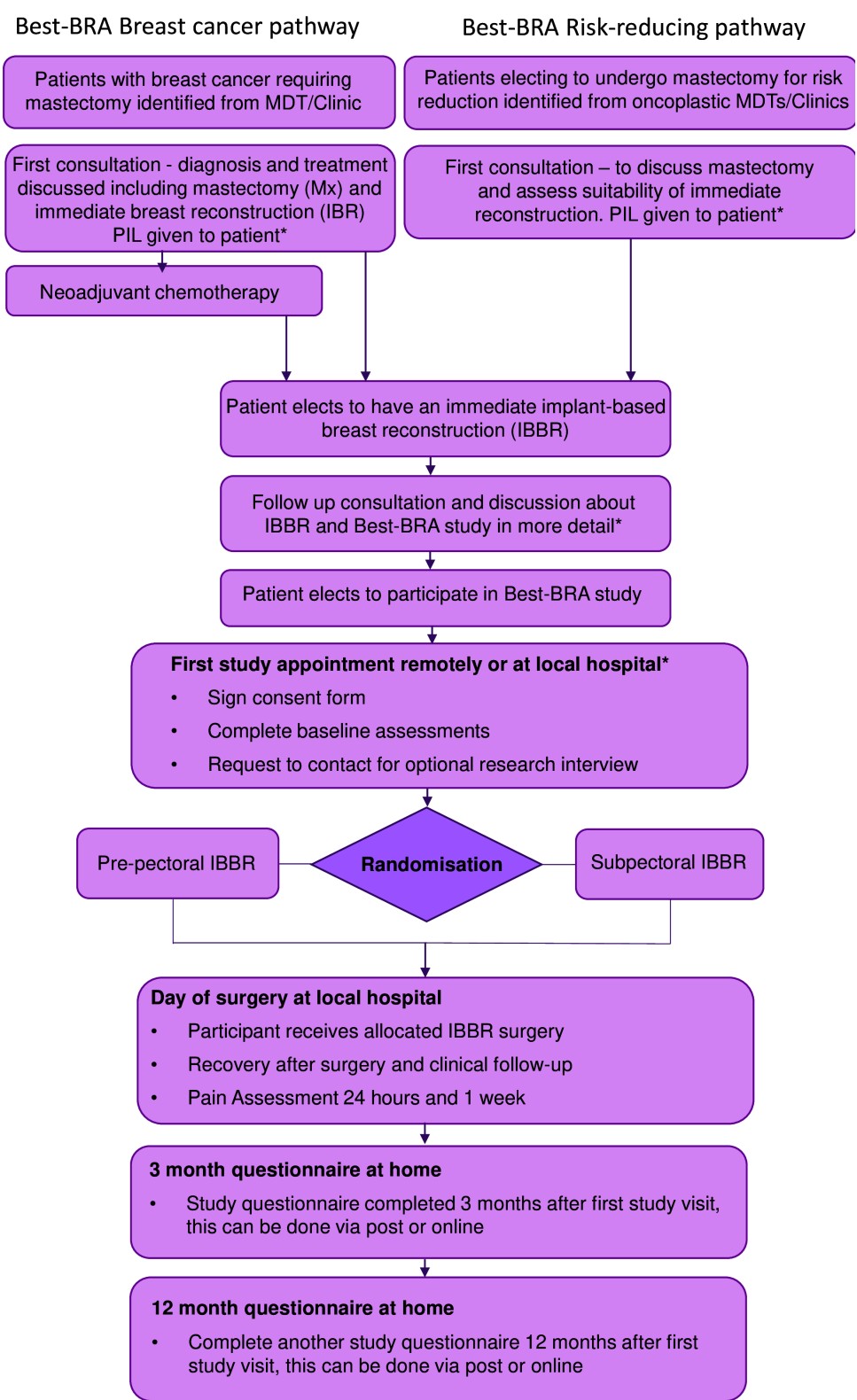

Figure 1 Best-BRA Study Schema. MDT, multidisciplinary team meeting; PIL, Participant Information Leaflet

local practice. Postoperative complications will be assessed using internationally agreed standardised definitions[61] by local clinical teams at routine postoperative hospital visits, that is, routine standard of care.

Clinically relevant complications assessed at 3 months and 12 months include:
► Implant loss defined as removal of the expander/implant without replacement.

- ► Infection requiring treatment with antibiotics and/or surgical debridement.
- ► Unplanned reoperation for complications relating to the implant reconstruction.
- ► Readmission to hospital for complications related to the implant reconstruction.

## Outcome measures

In the definitive trial, the proposed primary outcome will be women's satisfaction with the cosmetic outcome of their reconstruction at 12 months following surgery assessed using the 'Satisfaction with Breasts' domain of the validated BREAST-Q questionnaire.[62] On this basis, for the main trial, 100 patients will be required in each group to allow a true 0.5 SD difference in the BREAST-Q Satisfaction with Breasts domain scale to be detected at 5% significance and 90% power, assuming more than 85% of participants complete the questionnaire 12 months following surgery.

The purpose of this study is to determine the feasibility of conducting a future main trial, specifically whether it will be possible to recruit and randomise 200 patients undergoing IBBR and to collect the required data for the main trial.

The primary outcomes for the Best-BRA study are therefore

1. Recruitment (number of sites recruiting; proportion of eligible women approached that are randomised, women recruited per site per month).
2. Adherence to trial allocation.
3. Outcome completion rates at 3 and 12 months.

Secondary outcomes will be the feasibility of collecting the proposed primary and secondary outcomes for the main study: these will include:

1. Satisfaction with breasts using the validated BREAST-Q questionnaire at 12 months. We anticipate that this will be the primary outcome for the definitive trial.
2. Surgical complications, in particular implant loss, infection, readmission and reoperation within 3 and 12 months of random allocation.
3. The need for additional surgery to the reconstruction or the contralateral breast within 12 months of random allocation.
4. Pain scores assessed using a 10-point Likert scale at 24 hours and 1 week.
5. Objective panel assessment of cosmetic outcome at 12 months assessed using routinely collected patient photographs.
6. EQ-5D-5L health-related quality-of-life score[63] and ICECAP-A capability scores[64] at 1 year.
7. Other key patient-reported outcome domains included in the breast reconstruction core outcome set[65] including physical well-being (chest); emotional well-being; and animation assessed using appropriate subscales of the BREAST-Q.
8. The costs of prepectoral and subpectoral IBBR procedures.

## Data collection

Schedule of assessments is summarised in table 1. All data will be entered directly into electronic case report forms (CRFs) by local research teams and patients will complete electronic patient-reported outcome measures (PROMs) on REDCap.[66] The feasibility of uploading routinely taken anonymised preoperative and postoperative photographs directly onto the REDCap database following specific patient consent will be explored. Reminders will be sent to participants by email/text message for up to 4 weeks if questionnaires at 3 and 12 months have not been completed.

Participants can choose to discontinue their participation in the trial for any reason. With participant consent, research data obtained up to the point of discontinuation will be retained for analysis. Participants who decide to stop completing PROMs will continue to be followed up for complications and adverse event reporting through the review of their medical records unless they specifically object. Participants who are unwilling or unable to go ahead with their allocated treatment will be encouraged to continue in trial follow-up as per protocol and their alternative treatment recorded by the clinical staff team.

## Sample size calculation

A formal sample size calculation is not required for this external pilot study. At 12–18 months, the number of sites recruiting; proportion of eligible women that are approached and randomised; the number of women recruited per site per month will be reviewed and adherence to trial allocation will be reviewed. There are no plans to assess the feasibility of recruitment and randomisation in any specific patient subgroups. If the study has opened and is successfully recruiting at an acceptable number of sites with high adherence to treatment allocation and follow-up rates, a main trial will be considered feasible and parameters for the full study determined.

## Statistical analysis

Screening, recruitment, adherence to allocated procedure and completeness of primary outcome data will be presented by study group as a Consolidated Standards of Reporting Trials flow chart, overall and per centre. Secondary outcome measures will be presented as summary statistics at each assessment point. Summary statistics for the two study groups combined will be presented at the end of the pilot phase if the study continues into a definitive trial, with pilot data included in the definitive trial analysis.

## Health economics

The feasibility of applying a novel micro-costing framework to compare the relative costs of prepectoral and subpectoral IBBR will be explored.[67] A process map will be developed to track the patient pathway from first reconstruction consultation to last routine postoperative visit (usually up to 4 weeks following surgery). The process map will include a list of resources, equipment,

**Table 1** Schedule of assessment visits and outcomes measurements

| Timepoint | Baseline | Day of surgery | 24 hours | 1 week | 3 months | 12$^{+/-10}$ months |
|---|---|---|---|---|---|---|
| **Procedures** | | | | | | |
| Eligibility criteria review | ● | | | | | |
| Informed consent | ● | | | | | |
| Randomisation | ● | | | | | |
| Sociodemographics | ● | | | | | |
| Medical history | ● | | | | | |
| Drug history | ● | | | | | |
| Collection of routinely taken photographs | ● | | | | | ● |
| Pain scores | | | ● | ● | | |
| BREAST-Q | ● | | | | ● | ● |
| Operative assessments | | ● | | | | |
| EQ5D-5L and ICECAP-A | ● | | | | ● | ● |
| Postoperative histology MDT treatment decisions (eg, chemotherapy and radiotherapy) | | | | | ● | |
| **Safety** | | | | | | |
| Adverse events | | ● | | | ● | ● |

MDT, multidisciplinary team meeting.

consumables and implants involved in the procedures. Process mapping will be undertaken at several sites to ensure that all appropriate factors and patient pathways are considered. Resources that may potentially differ between the two procedures (eg, size of mesh; operative time; length of stay and number of postoperative visits) will be identified by reviewing the process maps and details collected on the CRFs. Data may also be gathered from review of electronic notes/medical records. The cost of resources used in the two procedures will be calculated using national tariffs (where available) or local costing estimates.

### QuinteT Recruitment Intervention (QRI)

A QRI will be embedded within the Best-BRA study to optimise recruitment and informed consent. The QRI will be implemented in two key phases—phase 1, in which sources of recruitment difficulties are rapidly investigated to inform the delivery in phase 2 of tailored interventions to optimise recruitment and informed consent. This will be supplemented with upfront pre-emptive recruitment training in study set up, tailored to anticipated issues and informed by previous QRIs.[42 52 55 68 69]

A multifaceted, flexible approach will be adopted in phase 1 using one or more of the following methods:

► Mapping of eligibility and recruitment pathways to collate basic data about the levels of eligibility and recruitment, and identify points at which patients opt in or out of the study.

► In-depth, semistructured interviews with a purposive sample of staff/site members involved with aspects of study design/management and recruitment across centres. Interviews will explore perspectives on the RCT and issues around recruitment, including views about the study design and protocol, treatment options, existing evidence and current practice. Interviews may also be undertaken with eligible patients to explore their views on the study, treatment options and provision of information. Interview topic guides will be used to ensure similar topic areas are covered across interviews, while still providing the scope for participants to raise issues of pertinence to them.

► Recording of consultations between healthcare staff and potentially eligible patients in which the study is discussed to explore information provision in relation to key study concepts and treatment options, recruitment techniques, engagement with patient treatment preferences and randomisation decisions to identify recruitment difficulties and improve information provision. Consultations will be listened to documenting instances such as unclear, insufficient or imbalanced information provision and unintentional transferring of clinician treatment preferences to patients, as well as aspects of good practice.

► Review of study documentation at set up and as recruitment progresses taking account of accumulating

interview and consultation data to ensure documents are clear and unbiased.

▶ Attendance at Trial Management Group (TMG) and investigator meetings to gain an overview of trial conduct and overarching challenges.

Interview and consultation recordings will be transcribed verbatim in full or in parts and edited to ensure anonymity of respondents. Data from phase 1 will be managed using qualitative data analysis software (such as NVivo). Interviews and recruitment consultations, along with screening logs and study documentation, will be subject to simple counts, content, thematic and targeted conversation analyses to identify aspects of recruitment that are causing difficulties within and across study sites (for further details, see [70]). Preliminary analysis will be used to inform strategies for phase 2 of the QRI and further data collection.

An account of the anonymised findings from all the data will be fed back to the Chief Investigator (CI). The QRI team, with the CI and TMG, will formulate a plan of action grounded in these findings to improve recruitment and information provision, with its format dependent on the nature of recruitment barriers identified (phase 2). Supportive and responsive group or individual feedback and training is likely to be a core component of the plan of action, including written recruitment 'tips' documents and suggested modifications to study pathways and patient-facing study material.

Phases 1 and 2 will be undertaken in an iterative and cyclical manner, continuing throughout the recruitment period with close monitoring of changes in screening log data and recruiter practice to optimise recruitment and informed consent, all in close collaboration with the CI and wider study team.

Study site/staff and potentially eligible patients will be provided with written information about the recording of consultations and interviews either as a stand-alone document (staff) or as part of the main PIL (patients) and invited to provide written consent. If the patient has yet to receive a PIL, a verbal explanation will be given and verbal consent sought initially at the start of the consultation, with the option to destroy recordings after receipt of PIL. Patient consent to the recordings/interviews is independent of their study participation decision. Data will be captured on encrypted devices and transferred, transcribed and stored in line with university data transfer and storage policies.

### Data management

Data will be entered onto the study REDCap database.[66] The system will incorporate data entry and validation rules to reduce data entry errors, and management functions to facilitate auditing and data quality assurance. The CI will have access to and act as custodian of the full dataset. Participants' data will be held securely held on the in line with data protection legislation. Personal data will not be kept for longer than is required for the purpose for which it has been acquired. Data will be held in compliance with the sponsor's standard operating procedures (SOPs). Anonymous data sets will be made 'open data' following publication and stored in the University of Bristol's Research Data storage facility.

### Monitoring, safety and audit

As this is a low-risk external pilot trial, comparing two procedures both of which are in routine clinical practice, the trial is overseen and audited by an independent joint Trial Steering/Data Monitoring Committee (TSC/DMC). The TSC/DMC will comprise an independent clinician, trials statistician and public and patient involvement (PPI) member. The TSC/DMC will meet once prior to recruitment of the first participant and convene at least annually to review adverse event data and any other ethical aspects that arise (see online supplemental appendix 3 for further details).

Research sites are responsible for reporting serious adverse events for their trial participants during the course of the trial on the electronic CRFs. The following adverse events are expected after surgery in this patient population:

▶ Anaesthetic complications, for example, stroke or cardiac events such as myocardial infarction.
▶ Return to theatre or readmission for complications of surgery.
▶ Wound infection requiring treatment with antibiotics, readmission or reoperation.
▶ Postoperative nausea and vomiting.
▶ Pain.
▶ Other infections (sepsis, septicaemia, abscess, respiratory).
▶ Other procedure specific complication including implant loss, mastectomy skin flap necrosis, infection, seroma.
▶ Allergic or anaphylactic reaction of Patent V blue dye used for sentinel node biopsy.

Adverse events will be documented and reported in accordance with the Sponsor's (North Bristol NHS Trust) Safety Reporting SOP.

### Protocol amendments

Any amendments to the protocol will be reported to the appropriate regulatory bodies, with a full copy of the current protocol available for download from the study website. There are no amendments to date.

### Ancillary and post-trial care

Participants will be treated according to standard care beyond their 12 months in the study.

### Public and patient involvement (PPI)

PPI members sit on the trial oversight committees and members of Independent Cancer Patients Voice have provided additional PPI input into all aspects of the study. PPI representatives have been actively involved in all stages of study design including the development of patient information leaflets and planning and refining recruitment strategies. Specifically, they have advised on how and when to

approach women undergoing IBBR to minimise the burden of participation and on the selection of outcome measures. PPI involvement will ensure that the patients' perspective is considered and remains the focus of the study. The PPI members will also assist in writing lay summaries and sharing findings of the study with a wider audience. They will advise on methods and content of communication with participants including newsletters and social media.

Please see online supplemental appendix 3 for further Administrative Information.

## Ethics and dissemination

The NHS Wales Research Ethics Committee 6 (20/WA/0338) reviewed and approved the study. Potential participants will be given at least 24 hours to decide whether they wish to take part and will provide written informed consent (either in person or electronically using the REDCap e-consent function) prior to entering the trial. Specific consent will be obtained for (1) upload of preoperative and postoperative clinical photographs; (2) future contact for long-term follow-up with PROMs or via routinely collected data to assess the long-term outcomes and costs of preoperative and subpectoral implant-based techniques and (3) the collection of anonymised clinical data about their care that will be kept indefinitely after the close of the study as publicly open data to support future research studies.

Results of the study will be presented at national and international meetings and published in peer reviewed journals. We will work with our PPI contributors to produce study summaries for patients. If the study is feasible, further funding will be sought for a full-scale trial comparing prepectoral and subpectoral implant-based reconstructive techniques.

### Author affiliations
[1]Population Health Sciences, University of Bristol Medical School, Bristol, UK
[2]Linda McCartney Centre, Royal Liverpool and Broadgreen University Hospitals NHS Trust, Liverpool, UK
[3]Department of Plastic Surgery, University Hospitals Coventry and Warwickshire NHS Trust, Coventry, UK
[4]Nottingham Breast Institute, Nottingham University Hospitals NHS Trust, Nottingham, UK
[5]Cancer Sciences Academic Unit, Faculty of Medicine, University of Southampton, Southampton, UK
[6]Breast Unit, Worcestershire Acute Hospitals NHS Trust, Worcester, UK
[7]Independent Cancer Patients' Voice, London, UK
[8]Bristol Breast Care Centre, North Bristol NHS Trust, Westbury on Trym, UK

**Acknowledgements** This study is being undertaken with the support of the NIHR Biomedical Research Centre at University Hospitals Bristol and Weston NHS Foundation Trust and the University of Bristol. This study was designed and delivered in collaboration with the Bristol Trials Centre (BTC), a UKCRC Registered Clinical Trials Unit in receipt of National Institute for Health Research CTU support funding. Trial steering and data monitoring committee members: Matthew Gardiner (chair), Dr Isabelle Smith (statistician) and Elizabeth Teasdale (PPI member).

**Contributors** SP conceived the study and obtained funding; KR, NM, CM, CC, AL, WH, JT, CH, JS, KF, LW, RIC, ST, SC, PF and SP contributed to the study design; CM provided statistical support; NM and CC led the QRI; KR is the Trial Manager responsible for the day-to-day running of the trial. KR prepared the first draft of this manuscript. All authors have read and approved the final manuscript.

**Funding** This work is funded by a National Institute for Health Research Clinician Scientist award to SP (CS2016-16-019).

**Disclaimer** The views expressed in this publication are those of the authors and not necessarily those of the NHS, the National Institute for Health Research or the Department of Health and Social Care.

**Competing interests** None declared.

**Patient consent for publication** Not applicable.

**Provenance and peer review** Not commissioned; externally peer reviewed.

**ORCID iDs**
Kirsty Roberts http://orcid.org/0000-0003-0765-3752
Chris Metcalfe http://orcid.org/0000-0001-8318-8907
Clare Clement http://orcid.org/0000-0002-5555-433X
Shelley Potter http://orcid.org/0000-0002-6977-312X

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
