## [Reviewer comments · BMJ Open]

ARTICLE DETAILS

TITLE (PROVISIONAL)	Best-BRA (Is subpectoral or pre-pectoral implant placement best in immediate breast reconstruction?) A protocol for a pilot randomised controlled trial of subpectoral versus pre-pectoral immediate implant-based breast reconstruction in women following mastectomy.
AUTHORS	Roberts, Kirsty; Mills, Nicola; Metcalfe, Chris; Lane, Athene; Clement, Clare; Hollingworth, William; Taylor, Jodi; Holcombe, Chris; Skillman, Joanna; Fairhurst, Katherine; Whisker, Lisa; Cutress, RI; Thrush, Steven; Fairbrother, Patricia; Potter, Shelley

VERSION 1 – REVIEW

REVIEWER	Hansson, Emma University of Gothenburg, Department of Plastic Surgery
REVIEW RETURNED	15-May-2021

GENERAL COMMENTS	This is a study protocol describing a trial randomizing patients to pre or subpectoral implant-based breast reconstruction. The authors should be commended for initiating this much needed trial. The main outcome is feasibility of randomization. The main weakness of the protocol is that inclusion and exclusion criteria and limitations in intervention are not detailed enough. Moreover, it should be stated if the authors intend to examine if randomization is feasible in any subgroups; for example, women of a particular age or sociodemographic status etc? • It should be clarified in the introduction that there is no strong scientific evidence that an immediate breast reconstruction (IBR) is better than a delayed breast reconstruction (DBR), or even that reconstruction is better than no reconstruction. The study the authors cite [3] conclude: ” The results indicate that breast reconstruction is not a universal panacea for the emotional and psychological consequences of mastectomy. Women still reported feeling conscious of altered body image 1 year postoperatively, regardless of whether or not they had elected breast reconstruction. Health professionals should be careful of assuming that breast reconstruction necessarily confers psychological benefits compared with mastectomy alone.”
---

The nice guidelines state that both IBR and DBR should be offered, and that the surgeon should be aware that some women do not want reconstruction at all (1.5).

- The statement that mesh/ADM is the standard of care should be clarified. There is no strong scientific evidence that mesh/ADM is superior to not using mesh/ADM. It should also be recognised that the British guidelines are under revision
<https://www.bapras.org.uk/media-government/news-and-views/view/statement-re-uk-use-of-biologic-or-synthetic-mesh-in-breast-surgery>

Methods

- Inclusion/exclusion criteria: The requirement that the patients have to be “considered eligible for both pre and subpectoral reconstruction by the surgical team” has to be clarified. The protocol should standardise which patients are considered eligible, and not leave it up to the local surgical team to decide, as patient selection is key when a pre-pectoral approach is used. For example, please specify required breast configuration (volume? ptosis?), mastectomy flap thickness (should fluorescence imaging be used to evaluate the mastectomy flaps?), smoking status, body mass index, comorbidity and medication, limitations in implant sizes and types used, limitations in types of meshes/ADMs, previous or future irradiation, and previous breast surgery. If it is not detailed, there will be too many confounding factors, making it very difficult to answer the research question. Limitations might also increase patient safety and make the two groups more homogenous.

- According to the protocol surgeons are encouraged to follow the Association of Breast Surgery (ABS) and British Association of Plastic, Reconstructive and Aesthetic Surgeons (BAPRAS)’ guidelines for the use of mesh. Please explain why surgeons are only encouraged to follow the guidelines. Do you plan to change this statement when the new guidelines are published?

- Power calculation: you state that no formal power calculation is needed. Nevertheless, it would be interesting to see sample size calculation for the definitive trial. You have performed some sort of calculation based on the satisfaction with breast domain of BREAST-Q, requiring a 0.5 standard deviation difference between the groups. Please consider performing the calculation based on the published distribution based minimal important differences (<https://pubmed.ncbi.nlm.nih.gov/31577663/>).

- Complication assessment: seromas requiring intervention should be included.

- Outcomes: please consider including animation as a separate outcome. You should also consider including oncologic factors, such as local recurrence, effects of radiation, ALCL etc.

- Adverse events: local recurrence, effects of radiation, and ALCL should also be included.

	 • Health economics: the cost during the first four weeks is not the most crucial factor in implant-based breast reconstruction, but rather the costs of long-term consequences, such as re-operations, corrections, conversion to autologous breast reconstruction etc. Therefore, cost difference between the two methods should be based on a longer time-period. • For the above-mentioned reasons, a longer follow-up than 12 months should be considered.
--	---

REVIEWER	Lee, Clara N The Ohio State University
REVIEW RETURNED	15-May-2021

GENERAL COMMENTS	This manuscript describes a feasibility study prior to a multi-site randomized controlled trial comparing two types of implant based breast reconstruction (IBBR). The paper is well written, and the study is well-designed. Some clarification or detail about methods for this study and the definitive trial is needed. The novel aspects of this study are its use of the “QuinteT Recruitment Intervention” (QRI) and plan to assess economic outcomes. However, there is some lack of basic detail about these approaches. I recommend giving a basic description of QRI early in the paper. In the methods section, more detail is needed about the qualitative analyses for the QRI study and about assessment of feasibility (see specific comments below). The introduction provides a strong description of the evidence to date and the reason the trial is needed. The background in the abstract and paper describe mesh-assisted subpectoral reconstruction as “standard of care”. It’s not entirely clear to me that this is the case. I recommend references or editing this description. In the United States, the Food and Drug Administration recently required specific counseling and documentation regarding the off-label nature of using biologic mesh for breast reconstruction, due to evidence of greater risk of complications. Since all patients in this trial will receive mesh, the authors may want to note these developments, for some context. The study setting is stated as “NHS breast and plastic surgery units in the UK”. More information is needed on which units, how many, and how they will be selected and included. All of that information has bearing on feasibility. Eligibility criteria are reasonable. One criterion is patients “who require a mastectomy”. I wasn’t sure what “require” means. Does this just mean “who will have a mastectomy”? If so, I would state it as such. If not, define “require”. The trial procedures are appropriate. They are generally described thoroughly. The choice of outcome measures is appropriate.
--

	Regarding enrollment- will clinicians be enrolling patients, or will research staff? It wasn't clear on page 8. A key part of the QRI method is audio recording the recruitment consultations. The methods section should describe how consent is obtained for this step and the data management for these audio recordings. If a patient is audio recorded but does not enroll in the study, what happens to their data, and how are those data kept secure? Are clinicians being recorded? And if so, will they be consented? Regarding feasibility assessment, the sample size calculation section states that "if the study is successfully recruiting at an acceptable number of sites with high adherence to treatment allocation and follow up rates", the trial will be considered. How are "acceptable number of sites" and "high adherence" and "high follow up rates" defined? The description of the qualitative analysis for the QRI is somewhat vague. How will the audio recordings and interviews be analyzed? For the economic analysis, "National and local unit costs will be used to estimate the incremental cost difference". I didn't understand what this means.
--	---

VERSION 1 – AUTHOR RESPONSE

Reviewer: 1: Dr. Emma Hansson, University of Gothenburg
Comments to the Author:

1. This is a study protocol describing a trial randomizing patients to pre or subpectoral implant-based breast reconstruction. The authors should be commended for initiating this much needed trial. The main outcome is feasibility of randomization.

We thank Reviewer 1 for this kind comment.

2. The main weakness of the protocol is that inclusion and exclusion criteria and limitations in intervention are not detailed enough. Moreover, it should be stated if the authors intend to examine if randomization is feasible in any subgroups; for example, women of a particular age or sociodemographic status etc?

Thank you for this comment. Best-BRA is designed as a pragmatic study, so the inclusion and exclusion criteria have been designed to reflect real-world practice and are deliberately broad.

We do not intend to evaluate the feasibility of randomisation in any particular subgroups, but rather in the trial population as a whole. We have added a statement to this effect in the 'Sample Size Calculation' section of the Methods.

Sample size calculation, page 14 para 1 now reads:

'At 12-18 months, the number of sites recruiting; proportion of eligible women that are approached and randomised; the number of women recruited per site per month will be reviewed and adherence to trial allocation will be reviewed. There are no plans to assess the feasibility of recruitment and randomisation in any specific patient subgroups.'

3. It should be clarified in the introduction that there is no strong scientific evidence that an immediate breast reconstruction (IBR) is better than a delayed breast reconstruction (DBR), or even that reconstruction is better than no reconstruction.

The study the authors cite [3] conclude: " The results indicate that breast reconstruction is not a universal panacea for the emotional and psychological consequences of mastectomy. Women still reported feeling conscious of altered body image 1 year postoperatively, regardless of whether or not they had elected breast reconstruction. Health professionals should be careful of assuming that breast reconstruction necessarily confers psychological benefits compared with mastectomy alone."

The nice guidelines state that both IBR and DBR should be offered, and that the surgeon should be aware that some women do not want reconstruction at all (1.5).

Many thanks for raising this issue. We agree that there is no strong scientific evidence that immediate breast reconstruction being better than delayed reconstruction. The focus of this paper, however, is on comparing different approaches to implant-based breast reconstruction. A detailed discussion of the evidence relating to the timing of breast reconstruction (or if reconstruction was better than no reconstruction) would therefore not be appropriate as this would detract the message of the introduction and make this section unnecessarily long.

We do, however, agree that this is an important point so have removed the word 'immediate' from the 2nd sentence of the introductory paragraph to reflect that breast reconstruction in general should be offered to women undergoing mastectomy. We feel that the use of the word 'offer' conveys that surgeons would be aware that not all women would choose to have a reconstruction but that they should be given the option.

4. The statement that mesh/ADM is the standard of care should be clarified. There is no strong scientific evidence that mesh/ADM is superior to not using mesh/ADM. It should also be recognised that the British guidelines are under revision <https://www.bapras.org.uk/media-government/news-and-views/view/statement-re-uk-use-of-biologic-or-synthetic-mesh-in-breast-surgery>

Thank you for requesting clarification regarding this. We agree that there is no robust evidence to support the safety or effectiveness of mesh-assisted breast reconstruction. The recently published updated UK ADM guidelines highlight this. We have edited the text to further emphasise the lack of evidence with specific reference to the updated guidelines. We have also included reference to the recent US FDA Safety Communication regarding the off label use of acellular dermal matrix (biological mesh) in breast reconstruction in response to Reviewer 2 (point 3 below)

Despite the lack of evidence, mesh assisted breast reconstruction has become the standard of care in the UK as evidenced by the UK iBRA study; a prospective cohort study of more than 2000 women undergoing immediate implant-based breast reconstruction between 2014 and 2016. Of these, more than two thirds of patients in the study had reconstructions performed with mesh. We have updated the text with this additional contextual information for clarify.

Introduction, para 2 page 5 now reads:

'There are, however, limited data to support the safety or effectiveness of mesh-assisted subpectoral techniques (15-19). The lack of high-quality evidence to support practice is highlighted in the recently updated UK mesh-assisted breast reconstruction guidelines (20) and in March 2021, the United States Food and Drug Administration (FDA) issued a Safety Communication stating that acellular dermal matrices, the most commonly used form of biological mesh, are not licenced for use in breast reconstruction and recommending careful discussion of the risks and benefits of mesh with women considering surgery (21). Despite this mesh-assisted procedures have become established as the standard of care in the UK, with two-thirds of the 2,108 patients undergoing IBBR between 2014 and 2016 in the UK iBRA multicentre prospective cohort study having mesh (18)'

5. Methods. Inclusion/exclusion criteria: The requirement that the patients have to be "considered eligible for both pre and subpectoral reconstruction by the surgical team" has to be clarified. The protocol should standardise which patients are considered eligible, and not leave it up to the local surgical team to decide, as patient selection is key when a pre-pectoral approach is used. For example, please specify required breast configuration (volume? ptosis?), mastectomy flap thickness (should fluorescence imaging be used to evaluate the mastectomy flaps?), smoking status, body mass index, comorbidity and medication, limitations in implant sizes and types used, limitations in types of meshes/ADMs, previous or future irradiation, and previous breast surgery. If it is not detailed, there will be too many confounding factors, making it very difficult to answer the research question. Limitations might also increase patient safety and make the two groups more homogenous.

Many thanks for this comment. The steering group has discussed the inclusion/exclusion criteria for the study at length. As per our response to point 2, Best-BRA has been designed as a pragmatic trial; i.e. one that aims to determine whether pre or subpectoral reconstruction is better in real-world practice. Eligibility for the Best-BRA study is therefore deliberately broad and based on the individual operating surgeon's assessment of whether the patient would be suitable for both pre or subpectoral techniques.

Restricting the inclusion/exclusion criteria would make this an explanatory trial. While we agree that this may improve safety, it would definitely not address the research question. We disagree that a pragmatic approach would result in too many confounding factors. We are recording information on known potential confounding variables but anticipate that the process of randomisation will lead to the creation of two comparable treatment groups (both in terms of known and unknown confounders) at baseline. This will allow the treatments to be compared fairly.

We have recommended that the participating surgeons consider the UK guidance for the best practice of implant-based reconstruction which suggests caution in smokers; patients with high body mass index; women who have received previous radiotherapy or in whom postmastectomy is likely to be needed and those with large or ptotic breasts. We know, however, from our earlier work in the iBRA study and ongoing work with the Pre-BRA pre-pectoral reconstruction study, that many surgeons offer implant-based procedures to a wide range of patients and the Best-BRA trial team have made a conscious decision that this should be an inclusive, pragmatic trial. We edited the text to ensure that the pragmatic nature of the planned study is highlighted.

6. According to the protocol surgeons are encouraged to follow the Association of Breast Surgery (ABS) and British Association of Plastic, Reconstructive and Aesthetic Surgeons (BAPRAS) guidelines for the use of mesh. Please explain why surgeons are only encouraged to follow the guidelines. Do you plan to change this statement when the new guidelines are published?

Many thanks for requesting clarification on this point. We are encouraging participating surgeons to follow the UK ABS and BAPRAS mesh guidelines to promote best practice in mesh-assisted implant reconstruction within the trial. Our work with the iBRA study (Potter et al 2019, Lancet Oncology) and in the ongoing Pre-BRA (Harvey et al 2020, BMJ Open) study shows that not all surgeons adhere to the guidelines. As this is a pragmatic study, we aim to generate 'real world' evidence of effectiveness so want to include all women being offered/considered suitable for both techniques technique. Restricting the inclusion criteria would make this an explanatory trial and limit the generalisability of the results.

The updated version of the ABS/BAPRAS guidelines has now been published. This reflects the latest evidence for the best practice of mesh-assisted implant reconstruction. We have updated the reference so that the latest version of the guidance is cited.

7. Power calculation: you state that no formal power calculation is needed. Nevertheless, it would be interesting to see sample size calculation for the definitive trial. You have performed some sort of calculation based on the satisfaction with breast domain of BREAST-Q, requiring a 0.5 standard deviation difference between the groups. Please consider performing the calculation based on the published distribution based minimal important differences (<https://pubmed.ncbi.nlm.nih.gov/31577663/>).

Many thanks for this comment. Looking ahead to the main Best-BRA study we extensively considered the published minimally clinical important difference (MCID) of 4 points. We decided against following this recommendation on the following basis:

Firstly, as the reviewer points out Voineskos et al's justification for a 4 point MCID is entirely statistical: 4 points on the Satisfaction with Breasts scale is 0.2 SD. There is no consideration of the patient view on this, and no comparison with other types of measure (see <https://link.springer.com/article/10.1186/s13063-014-0526-8#Tab1> for other approaches to determining the MCID). We do not see that recommendations based solely on the distribution approach can be reliable. Our choice of a MCID of 0.5 SD is also not based on any empirical evidence, but we don't agree that in the absence of evidence such a small MCID as 0.2 SD should be assumed.

Voineskos et al only superficially consider the implications of such a small MCID for the cost and timely delivery of a randomised controlled trial. Repeating the calculation, we present in our manuscript with a MCID of 4 points gives a target sample size total of 1052 patients for a two-group trial. This is considerably larger than any other trial in this clinical area, for example the BRIOS trial published in Lancet Oncology in 2018 used the BREAST-Q Satisfaction with Breasts Scale in a calculation that assumed a MCID of 10 points and 80% power and indicated a target sample size of 140 patients. The danger with attempting to deliver larger trials to detect differences in outcome that are smaller than the true MCID is that they won't be delivered before clinical practice has moved on, without evidence, to the point where a randomised evaluation is no longer possible due to lack of equipoise amongst surgeons.

The Satisfaction with Breasts scale has 15 items with 4 possible responses for each item. Taking the simplest approach (to allow the MCID to be understood in terms of responses to individual items) by scoring these responses to each item 0 to 3, and adding up the 15 items in the scale, gives a range of 0 to 45. This is very approximately doubled to give the actual scoring range of 0 to 100, hence 4 points on the actual scoring range is about 2 points on our simplified scale. This indicates that the MCID of 4 points corresponds to an identical response to 13 out of 15 items and responding "somewhat satisfied" rather than "somewhat dissatisfied" on the remaining two items. Accepting this will be an important difference to individual women affected, at the group level considered in a randomised trial, this is considered a small difference (requiring 1052 participants to distinguish this true difference from random variation). The 0.5

SD difference used by BRIOS and ourselves corresponds to identical responses on 10 out of 15 items and responding "somewhat satisfied" rather than "somewhat dissatisfied" on five items, which we expect will be widely considered as a clinically important difference.

Furthermore, as the reviewer suggests, it is interesting to look ahead to the likely size of the main study as it is not useful to conduct a pilot study for a main study that we already know would be undeliverable. However, at this stage it is exactly that, a look ahead. The choice of MCID for the main study will need to be discussed with patients, the clinical community and funders before proceeding with the next stage of the research.

8. Complication assessment: seromas requiring intervention should be included.

We are measuring surgical complications as a key secondary outcome (see page 13, secondary outcomes point ii.). This will include seromas requiring intervention.

9. Outcomes: please consider including animation as a separate outcome. You should also consider including oncologic factors, such as local recurrence, effects of radiation, ALCL etc.

We are already planning to include an assessment of animation using the newly developed BREAST-Q animation scale (see page 13, point viii. 'Other key patient reported outcomes'). We will also include effect of radiation BREAST-Q scale as part of this assessment.

We appreciate that oncological outcomes are important but the primary endpoint for the main trial will be assessed at 12 months. It is unlikely that patients will have developed ALCL or local recurrence at this time point. Furthermore, we will be recruiting patients having mastectomy for risk-reduction as well as breast cancer. As the sample size for the main trial will be relatively small (approx 200-300) and it is anticipated that approximately 25-30% of included patients will be having risk reducing surgery, the study group did not think that oncological outcomes should be specifically included the study.

10. Adverse events: local recurrence, effects of radiation, and ALCL should also be included.

We thank Reviewer 1 for this suggestion. As per our response above, the steering group considered the definitions of adverse events in collaboration with the study sponsor when designing the study. ALCL, although unlikely in the 1st 12 months following surgery would be captured within 'procedure specific complications'. We would not consider effects of radiation to be an adverse event but will collect it as a patient reported secondary outcome. Local recurrence would not be considered an 'adverse event' of the surgery based on our discussion with the study sponsor. We will collect oncological and adjuvant treatment data as part of the main study.

11. Health economics: the cost during the first four weeks is not the most crucial factor in implant-based breast reconstruction, but rather the costs of long-term consequences, such as re-operations, corrections, conversion to autologous breast reconstruction etc. Therefore, cost difference between the two methods should be based on a longer time-period.

Thank you for this comment. We agree that the cost differences between the two procedures are likely to manifest beyond 12 months in terms of revisional surgery. A trial with extended follow up, however, is unlikely to be funded in the current resource constrained funding climate so we have pragmatically decided on a 12 month follow up in keeping with the core measurement set for implant-based breast reconstruction. We will, however, obtain consent to contact patients for long term follow up with patient

reported outcomes and via routinely collected data (such as Hospital Episode Statistics, HES) to allow the long-term outcomes and costs of pre and subpectoral reconstruction to be assessed. We have edited the 'Ethics and Dissemination' to provide clarity on this point.

Ethics and Dissemination, page 20 para 1 now reads

... 'Specific consent will be obtained for i) upload of pre and postoperative clinical photographs; ii) future contact for long-term follow up with PROMs or via routinely collected data to assess the long-term outcomes and costs of pre and subpectoral implant-based techniques'

12. For the above-mentioned reasons, a longer follow-up than 12 months should be considered.

We agree but have decided on a 12 month follow up in the first instance to increase the likelihood of successfully securing funding for a main trial. We will, however, consent patients to be contacted so that the long-term outcomes and costs of pre and subpectoral reconstruction can be explored and have revised the text accordingly (see response to point 11 above).

Reviewer: 2: Dr. Clara N Lee, The Ohio State University

Comments to the Author:

1. This manuscript describes a feasibility study prior to a multi-site randomized controlled trial comparing two types of implant-based breast reconstruction (IBBR). The paper is well written, and the study is well-designed. Some clarification or detail about methods for this study and the definitive trial is needed.

We thank Reviewer 2 for their kind comment and will attempt to address the comments below.

2. The novel aspects of this study are its use of the "QuinteT Recruitment Intervention" (QRI) and plan to assess economic outcomes. However, there is some lack of basic detail about these approaches. I recommend giving a basic description of QRI early in the paper. In the methods section, more detail is needed about the qualitative analyses for the QRI study and about assessment of feasibility (see specific comments below).

Thank you for pointing this out. A basic description of the QRI has now been provided in the introduction.

Introduction, page 7 para 4 now reads:

'We therefore aimed to undertake an external pilot RCT (Best-BRA study- is subpectoral or pre-pectoral implant placement Best in immediate BReAst reconstruction?). In anticipation of recruitment challenges a QuinteT Recruitment Intervention (QRI) – a complex intervention that aims to rapidly identify, understand, and address sources of recruitment difficulties (54) – has been embedded in the study protocol. Having previously been applied to over 60 RCTs to date, the QRI has led to insights about recruitment issues and the development of targeted strategies that have facilitated successful completion of surgical trials that had previously been considered impossible (54, 55).'

Information about the methods of analysing QRI data has also been provided in the methods section.

Methods, page 16 para 2 now reads

'Interview and consultation recordings will be transcribed verbatim in full or in parts and edited to ensure anonymity of respondents. Data from Phase 1 will be managed using qualitative data analysis software

(such as NVivo). Interviews and recruitment consultations, along with screening logs and study documentation, will be subject to simple counts, content, thematic and targeted conversation analyses to identify aspects of recruitment that are causing difficulties within and across study sites (for further details see ref 70). Preliminary analysis will be used to inform strategies for Phase 2 of the QRI and further data collection.'

3. The introduction provides a strong description of the evidence to date and the reason the trial is needed. The background in the abstract and paper describe mesh-assisted subpectoral reconstruction as "standard of care". It's not entirely clear to me that this is the case. I recommend references or editing this description. In the United States, the Food and Drug Administration recently required specific counselling and documentation regarding the off-label nature of using biologic mesh for breast reconstruction, due to evidence of greater risk of complications. Since all patients in this trial will receive mesh, the authors may want to note these developments, for some context.

Many thanks for requesting further information regarding this. As per our response to Reviewer 1, point 4 above, we have revised the text to highlight the recently updated mesh-assisted breast reconstruction guidelines and FDA safety warning regarding the use of mesh. Despite the limited evidence, mesh has become the standard of care in the UK and the majority of implant-based reconstructions performed in the UK are now single-stage mesh assisted procedures. We have included further information regarding participants in the UK iBRA multicentre prospective cohort study to illustrate this.

Introduction, para 2 page 5 now reads:

'There are, however, limited data to support the safety or effectiveness of mesh-assisted subpectoral techniques (15-19). The lack of high-quality evidence to support practice is highlighted in the recently updated UK mesh-assisted breast reconstruction guidelines (20) and in March 2021, the United States Food and Drug Administration (FDA) issued a Safety Communication stating that acellular dermal matrices, the most commonly used form of biological mesh, are not licenced for use in breast reconstruction and recommending careful discussion of the risks and benefits of mesh with women considering surgery (21). Despite this mesh-assisted procedures have become established as the standard of care in the UK, with two-thirds of the 2,108 patients undergoing IBBR between 2014 and 2016 in the UK iBRA multicentre prospective cohort study having mesh (18)'

4. The study setting is stated as "NHS breast and plastic surgery units in the UK". More information is needed on which units, how many, and how they will be selected and included. All of that information has bearing on feasibility.

We thank Reviewer 2 for this comment. This is a pragmatic trial and all NHS breast and plastic units performing pre- and subpectoral implant-based breast reconstruction are eligible to participate in the study. They will be included based on local surgeons' desire to participate and the local centre's research capacity. There are no other selection criteria for participating units. This is hypothesised to be a 'difficult to do trial' so a key aspect of feasibility is to open as many centres as possible during the 12-18 month recruitment period. We have added additional detail to highlight that participating units need to be able to offer both pre and subpectoral implant-based reconstruction techniques.

Study setting, page 9 para 1 now reads

'NHS breast and plastic surgery units in the UK offering both subpectoral and pre-pectoral immediate implant-based breast reconstruction to women following mastectomy'.

5. Eligibility criteria are reasonable. One criterion is patients "who require a mastectomy". I wasn't sure what "require" means. Does this just mean "who will have a mastectomy"? If so, I would state it as such. If not, define "require".

Many thanks for pointing out this error. We have revised the text as suggested.

6. The trial procedures are appropriate. They are generally described thoroughly. The choice of outcome measures is appropriate.

Thank you

7. Regarding enrolment- will clinicians be enrolling patients, or will research staff? It wasn't clear on page 8.

Many thanks for requesting clarification on this point. We have updated the text to reflect that surgeons and research teams will be involved in recruiting and randomising patients in the study. Treating surgeons will introduce the study and confirm eligibility prior to randomisation.

Recruitment, page 9 para 2 now reads:

'Participants will be recruited and randomised over a 12-18 month period by surgeons and research teams working together at participating centres. All female patients undergoing mastectomy who elect to have an IBBR will form the target population. Research teams at each site will maintain a trial screening log'

Page 10 para 3 now reads:

'Participants will be randomised by the local research team after eligibility and consent have been confirmed at the final clinic visit by the treating surgeon prior to admission for surgery

8. A key part of the QRI method is audio recording the recruitment consultations. The methods section should describe how consent is obtained for this step and the data management for these audio recordings. If a patient is audio recorded but does not enrol in the study, what happens to their data, and how are those data kept secure? Are clinicians being recorded? And if so, will they be consented?

Thank you for asking for additional clarification. Information has now been added to the methods section relating to consent for the consultation recordings (and interviews) and data storage.

Methods, page 17, para 4 now reads:

'Study site/staff and potentially eligible patients will be provided with written information about the recording of consultations and interviews either as a stand-alone document (staff) or as part of the main PIL (patients) and invited to provide written consent. If the patient has yet to receive a PIL, a verbal explanation will be given and verbal consent sought initially at the start of the consultation, with the option to destroy recordings after receipt of PIL. Patient consent to the recordings/interviews is independent of

their study participation decision. Data will be captured on encrypted devices and transferred, transcribed and stored in line with university data transfer and storage policies’.

9. Regarding feasibility assessment, the sample size calculation section states that “if the study is successfully recruiting at an acceptable number of sites with high adherence to treatment allocation and follow up rates”, the trial will be considered. How are “acceptable number of sites” and “high adherence” and “high follow up rates” defined?

We have specifically not pre-specified any of these variables as we were concerned that this may limit or restrict our definition of feasibility. The aim of the feasibility phase is to set up as many sites as possible and determine whether it is possible to recruit patients from these sites. Progress will be reviewed at regular intervals during the recruitment phase by the study steering group and the independent data monitoring committee. The number of sites able to open and recruit during the feasibility phase will be assessed against the target sample size for the main trial to determine whether it will be possible to undertake a main trial that will complete accrual over an acceptable time period.

10. The description of the qualitative analysis for the QRI is somewhat vague. How will the audio recordings and interviews be analyzed?

More information about analysis of QRI has been given in the methods p16 (as described above). As the QRI uses various methods of analysis, we have referred the reader to a new reference for a detailed explanation of how these analytical approaches are undertaken within the QRI.

11. For the economic analysis, “National and local unit costs will be used to estimate the incremental cost difference”. I didn’t understand what this means.

Apologies that this is unclear. We have rephrased for clarity.

Health Economics, page 15, para 1 now reads:

‘The cost of resources used in the two procedures will be calculated using national tariffs (where available) or local costing estimates.’

We note that we have slightly exceeded the word limit for the manuscript however this is reflective of the increased details requested by Reviewer 2 concerning the methods section for the QRI.

VERSION 2 – REVIEW

REVIEWER	Hansson, Emma University of Gothenburg, Department of Plastic Surgery
REVIEW RETURNED	01-Sep-2021

GENERAL COMMENTS	The authors have adressed the issues raised previously and it is now considerably clearer and easier to see what the authors are trying to achieve.
---

REVIEWER	Lee, Clara N
-----------------	--------------

	The Ohio State University
REVIEW RETURNED	03-Sep-2021

GENERAL COMMENTS	The authors have thoroughly addressed all reviewer comments.
--